ecology/evolution

phenotypic plasticity, antipredator response, lifespan, longevity, kairomone, senescence

**Author for correspondence:**
Barbara Pietrzak
e-mail: b.pietrzak@uw.edu.pl

# Phenotypic plasticity of senescence in *Daphnia* under predation impact: no ageing acceleration when the perceived risk decreases with age

Barbara Pietrzak[1], Max Rabus[2], Maciej Religa[1], Christian Laforsch[2] and Maciej J. Dańko[3]

[1]Department of Hydrobiology, Institute of Zoology, Faculty of Biology, Biological and Chemical Research Centre, University of Warsaw, Zwirki i Wigury 101, 02-089 Warszawa, Poland
[2]Animal Ecology I and BayCEER, University of Bayreuth, Universitaetsstraße 30, 95447 Bayreuth, Germany
[3]Max Planck Institute for Demographic Research, Konrad-Zuse-Straße 1, 18057 Rostock, Germany

BP, 0000-0001-5446-6277

Recognising the nature of the predation risk, and responding to it accurately, is crucial to fitness. Yet, even the most accurate adaptive responses to predation risk usually entail costs, both immediate and lifelong. Rooting in life-history theory, we hypothesize that an animal can perceive the nuances of prey size and age selectivity by the predator and modulate its life history accordingly. We test the prediction that—contrary to the faster or earlier senescence under predation risk that increases with prey size and age—under predation risk that decreases with prey size and age either no senescence acceleration or even its deceleration is to be observed. We use two species of indeterminate growers, small crustaceans of the genus *Daphnia*, *Daphnia Pulex* and *Daphnia magna*, as the model prey, and their respective gape-limited invertebrate predators, a dipteran, midge larva *Chaoborus flavicans*, and a notostracan, tadpole shrimp *Triops cancriformis*. We analyse age-specific survival, mortality and fertility rates, and find no senescence acceleration, as predicted. With this study, we complete the picture of the expected non-consumptive phenotypic effects of perceived predation pressure of different age-dependence patterns.

# 1. Introduction

An accurate response to environmental challenges is crucial for an organism to maximize its fitness. It includes both recognising the actual threat or opportunity and modifying the phenotype accordingly. These modifications often entail costs that need to be taken into account. Hiding in refuge or forming an armour may increase survival under predation risk but may also reduce the number of resources available for other crucial aspects of life such as reproduction and maintenance. The reduction may be both direct or indirect, the latter via shorter time spent on foraging or via physiological constraints.

Adaptive responses to predation risk, and their immediate costs, have received considerable attention in research [1–12]. What is far less studied are lifelong responses to perceived predation risk. The basic rationale for that lack of study coverage for senescent individuals is the assumption that under predation, the life of the prey is shorter, and the relative contribution of the late-life age classes to fitness is lower than when a predator is absent. Another reason is that it is difficult to study senescence and other late-life-history traits in the wild where only a few individuals survive to the old ages [13,14]. A strong predation impact is associated with shorter life expectancy in the affected population. Yet, those particular individuals that do survive, are not necessarily short-lived by 'genetic design', especially in cases where the young adults are more vulnerable to predation.

The idea of extrinsic mortality-driven decline in the force of selection with age, formalized later by Hamilton [15], inspired Williams to formulate the expectation that 'low adult death rates should be associated with low rates of senescence, and high adult death rates with high rates of senescence' [16]. This was later specified: increase of extrinsic mortality with age leads to potential lifespan shortening via increased rates of senescence [17–20]. The initial idea thus laid the groundwork for theories of ageing, including probably the most accepted, the disposable soma life-history theory [21,22], where senescence is a by-product of life-history trade-offs. Particularly, it has been suggested that under predation risk, shortened lifespan is an expected cost of increased early life investments in reproduction and decreased allocation to soma growth, maintenance and repair [19,23].

The majority of theoretical work is based on the assumption that extrinsic mortality is independent of age and size (discussed in [18,19]), despite age-independent predator mortality being rather rare in nature [17]. Indeed, predators are typically selective, and 'it is hard to conceive of a predator in nature without some elements of age specificity in prey choice' [24]. They often prey preferably on e.g. younger or older, as weaker, or either bigger or smaller individuals (e.g. [25–30]). One example is the predator preference that increases with the size of indeterminately growing prey (e.g. [31,32]). In the presence of a predator selectively feeding on larger individuals, the chance of survival decreases with size, and consequently also with age. Under such an extrinsic mortality risk, investing both in growth and in somatic maintenance and repairs is futile; hence, an earlier or accelerated senescence is promoted [33]. In such a case, a shortened physiological lifespan may be a negligible cost not manifested in reduced fitness and possibly traded-off for early fitness components.

The opposite situation occurs when predation risk decreases with size and/or age, which is also a frequent case. Juveniles and young adults are often more susceptible to predators than older individuals, which by gaining experience or simply growing larger escape the risk. Still, they bear costs of living in stress and, most often, of enabling other defences, such as a strengthened armour [34]. Yet, under such predation mode, the fitness cost of shortened physiological lifespan may be non-negligible and not easy to predict. Predation *per se* decreases the chances of survival to older ages, even if the decrease becomes ever smaller with age. On one hand, both living in stress by itself and investing in morphological defences or in faster growth to escape predation incur costs, so as above, senescence may occur earlier. On the other hand, via more intensive growth and bigger size, an animal may reach higher temporal reproductive rates later in life, pushing away the 'selection shadow'. Furthermore, in organisms that experience indeterminate growth, the growth mechanism may work as a tissue renewal process, at least partially stopping the senescence [35,36].

Rooting in the life-history theory, we hypothesize that an animal can perceive the age- and size-specific nuances of the threat involved and modulate its life history accordingly. We set out to test the prediction that—contrary to the faster or earlier senescence under predation risk increasing with prey age [33,37]—under predation risk that decreases with prey age either no senescence acceleration or even its deceleration is to be observed (table 1). We expect this to be particularly valid where older prey contributes more to the reproduction, like in the case of many indeterminate growers, and where trading-off long life easily for early investments might not be favoured.

**Table 1.** The hypothesis: the expected non-consumptive phenotypic effects of perceived predation pressure of different age-dependence patterns.

| predation pressure | model prey–predator | growth | early reproductive effort | senescence | lifespan | support references |
|---|---|---|---|---|---|---|
| increasing with age/size | planktonic crustacean | | increased | | | Law [24] |
| | versus fish | decelerated, levelling off | | earlier/faster | shorter | Taylor & Gabriel [38] |
| | | | | | | Pietrzak *et al.* [33] |
| decreasing with age/size | planktonic crustacean | | increased or decreased | | | Law [24] |
| | versus gape-limited | accelerated, continuous | | | | Taylor & Gabriel [38] |
| | invertebrate | | | decelerated or no effect | longer or no effect | this study |

Some evidence exists that animals not only perceive threats but also anticipate the risk of dying in the future. Bluebirds discriminate juvenile versus adult predation risk and respond with their reproductive behaviour accordingly [39]. The microcrustacean *Daphnia*, an important model organism in ecology, evolution and environmental science, is perfectly suited to study similar hypotheses owing to its pronounced plasticity in morphology and life history. For instance, *Daphnia* discriminates between different species of predators, each exerting different selective pressure demanding an adequate phenotypic response [3,4,40–45]. It is not known, however, if *Daphnia* can discriminate between predation risk that decreases and one that increases with age. Nor are there studies directly addressing the phenomenon of an organism modulating its life history, senescence included, according to the age-dependence of the threat perceived. It has been suggested, though, that *Daphnia* may behaviourally control its life history, including longevity, via thermal and depth preference [46].

## 2. Methods

### 2.1. Study organisms

We use a well-described prey–predator system of *Daphnia* versus gape-limited invertebrate predator in two specific settings. In such a system, juveniles and young adults are particularly vulnerable, while larger prey— older and structurally defended individuals—can escape predation pressure. The planktonic cladoceran *Daphnia* is a well-suited model to test our predictions owing to several characteristics, including (i) indeterminate growth, producing tight link between age and size, (ii) cyclical parthenogenesis, enabling testing dozens of genetically identical individuals under different environmental settings, and (iii) existence of massive body of literature on its ecology, prey–predator interactions in particular, enabling a well-informed choice of the particular system. We choose two particular two-species systems.

One is *Daphnia pulex* versus a dipteran, midge larva *Chaoborus flavicans*. *Daphnia pulex* exposed to chemical cues released by *Chaoborus* predation grow larger, form a longer tail spine and protective neck teeth, develop a stronger armour, delay reproduction and/or produce fewer and larger offspring [34,47,48].

The other is *Daphnia magna* versus a notostracan, tadpole shrimp *Triops cancriformis*. *Daphnia magna* exposed to cues of this predator expresses a bulky body shape and fortifies its exoskeleton, both enabling escape upon capture [49–51].

### 2.2. *Chaoborus* experiment

The experimental *D. pulex* clonal lineages originated from a small temporary pond in the valley of the Vistula river in the Wawer district of Warsaw, Poland (W), and a flow-through pond on Czarna Struga, a small tributary of Roś lake in Mazury, Poland (Cz). *Chaoborus* larvae were collected from a permanent pond in the Ujazdowski Park in Warsaw.

Each *D. pulex* clonal line was started with a single individual taken from clonal cultures of the Department of Hydrobiology, University of Warsaw. The selected clones were further cultured in the laboratory for about a month before starting the experiment, at a constant temperature of 20°C, summer photoperiod (16 L : 8 D), and in 1 l glass vessels filled with filtered and aerated lake water. The animals were fed daily with the suspension of green algae *Acutodesmus obliquus* at a concentration corresponding to $1 \, \mathrm{mg} \, C_{org} \, l^{-1}$. The water was changed every other day, and the cultures were maintained under constantly controlled conditions to remove the maternal effect. Each time, individuals from the second clutch were taken into either further culture or, finally, the experiment.

A total of 140 individuals of *D. pulex* were used in the experiment (2 clones × 2 treatments × 7 vessels × 5 individuals). Seventy newly born individuals of each clone were randomly distributed between fourteen 0.3 l glass vessels: seven with control and seven with kairomone water. Throughout the experiment, the animals were kept in 50 ml medium ind.$^{-1}$, starting with five individuals in 250 ml medium, and after each death, the amount of medium per vessel was reduced accordingly.

Kairomone water was obtained by incubating *Chaoborus* larvae at a density of 25 ind. $l^{-1}$ for 48 h, at 4°C to prevent quick development and metamorphosis during the experiment and fed with *D. pulex*. To maintain cue quality, all these parameters, i.e. temperature, predator density and diet, were kept constant throughout the experiment. Apart from predator larvae exposition, both control and kairomone medium were obtained in the same way where pre-treated lake water (pre-filtered and aerated) was filtered (both prior and post exposition) through the ceramic filter of pore size of 0.3 μm. The medium was changed by

transferring the remaining *D. pulex* to new vessels filled with the appropriate amount of the appropriate medium at 20°C, containing a freshly added suspension of *A. obliquus* in a concentration corresponding to $1 \, \mathrm{mg} \, C_{\mathrm{org}} \, l^{-1}$.

Live and dead individuals were counted, the offspring counted and removed, the medium was changed and the animals fed daily. The experiment was finished soon after the size of each experimental cohort was reduced at least by half.

## 2.3. *Triops* experiment

The experimental *D. magna* clonal lineage originated from a former fishpond north of Munich, Germany (K34 J). *Triops cancriformis* came from a laboratory cultured clonal line from the University of Vienna.

The experimental procedure has been described in detail by Rabus & Laforsch [48]. In brief, the experiment was conducted in a climate chamber at $20 \pm 0.5°C$ under fluorescent light at summer photoperiod (15 L : 9 D), in glass aquaria (30 × 20 × 20 cm) filled with 10 l of artificial medium. Age-synchronized cohorts of *Daphnia* were thus cultured for three generations, at a constant density of 50 individuals per aquarium, either in control medium or in the direct contact with one *T. cancriformis* of 20 mm body length. They were fed daily with the suspension of green algae *A. obliquus* at a concentration corresponding to $1 \, \mathrm{mg} \, C_{\mathrm{org}} \, l^{-1}$.

When the third generation released their first clutch, 40 randomly chosen neonate individuals of *D. magna* were used in the experiment (2 treatments × 5 replicate aquaria × 4 individuals). The individuals were randomly distributed between 40 labelled 50 ml glass vessels closed with a gauze cap (mesh size: 0.4 mm). The vessels were then distributed between the 10 aquaria, where current produced by an air stone guaranteed a constant exchange of medium, predator cues and algae inside the vessels.

Three *T. cancriformis*, with a body length over 20 mm, were placed into each aquarium of the induction treatment. *Daphnia* was fed daily by adding *A. obliquus* at a concentration corresponding to $0.7 \, \mathrm{mg} \, C_{\mathrm{org}} \, l^{-1}$. *Triops cancriformis* were fed with 10 pellets of fish food and to induce a stronger response in experimental *D. magna*, with an addition of 10 freshly killed *D. magna* of the same clone [52]. The fish food and killed *D. magna* were added daily to both the induction and the control treatment. The aquaria were cleaned daily of exuviae, faeces, fish food remnants and dead *Daphnia*. Medium and aquaria were exchanged, and the glasses and the air stones cleaned, every 4 days. In addition, to maintain a constant level of kairomones during the experiment, medium conditioned with *T. cancriformis* cues was used, obtained by exposing three individuals in 30 l of medium for 1 day, diet as above and constant throughout the experiment.

*Daphnia magna* were checked and offspring counted and removed twice per day. The experiment was finished when all experimental individuals died.

## 2.4. Data analysis

All analyses were performed in the R language and environment [53]. The survivorship functions were plotted and analysed by a Kaplan–Meier estimator of survival [54] using the *survival* package [55,56]. The age-specific mortality rates and fertility rates were analysed by generalized additive models (GAM) using the *mgcv* package [57]. Every fertility or mortality model assumed treatment (control or predator) as categorical predictor, age as a smooth term (s), tensor product interaction between age and treatment (ti), and offset equal to the log of exposures (individual days lived). As dependent variables, we used offspring counts for fertility and death counts for mortality. To avoid over-dispersion problems, we used negative binomial distribution. The model selection was performed by adding an extra penalty to each term so that it could be penalized to zero. All models were fitted using restricted maximum likelihood. In addition to this general procedure, there were substantial differences in the calculation of exposures and counts among different models and types of data.

Data for *Chaoborus* experiments were collected daily in five individual vessels, which were affected by right censoring events (experiment ceased when some individuals were still alive in some of the vessels). This censoring was included in the calculation of death counts and exposures. The effect of the vessel was investigated using generalized additive mixed models, where a vessel was treated as random effects and modelled as simple smooths (re smooth basis). Because the effect of the vessels was always non-significant, random effects were not included in the final models. We assumed that each event (birth or death) occurred in the middle of the age interval.

Data for *Triops* were collected for every individual at every clutch, so a particular clutch was reached in different ages per individual. Individuals who experienced accidental deaths were treated as right

censored. The death and censoring were assumed to occur in the middle of the last age class. The length of the last age class was assumed to be the same as the length of an age class of the last clutch. Before the mortality model was fitted, the individual mortality data were binned into one-day intervals (as a state of being alive is independent on interval lengths before death). Fertility data were analysed on an individual basis assuming that fertility events occur at the end of the clutch-age class interval.

The overall fertility rates for both kinds of data were analysed using generalized linear model Poisson regressions [58]. The offset was equal to the log of sum of the exposures for each treatment. It was not necessary to control for over-dispersion, so basic Poisson distribution was assumed.

# 3. Results

## 3.1. *Chaoborus* experiment

There was no significant effect of the presence of *Chaoborus* kairomone on survival (figure 1*a*) and mortality (figure 1*b* and table 2) in *D. pulex*, and age-specific mortality rates increased roughly exponentially with age in both treatments. The fertility rates seemed to be higher in *Chaoborus* treatment only at later ages (interaction of treatment with age), but the effect was only close to significance (figure 1*c* and table 3). Aggregated fertility data revealed an 11% higher overall fertility rate in kairomone than in control treatment ($z = 5.45$, $p < 0.0001$; figure 1*d*).

## 3.2. *Triops* experiment

There was no significant effect of the presence of *Triops* kairomone on survival (figure 2*a*) and mortality (figure 2*b* and table 4) in *D. magna*. The age-specific mortality rates increased exponentially with age and seemed to have identical slopes (Gompertz's ageing rates) in both treatments. The fertility rate was significantly higher in kairomone than in control treatment (table 5). The significant interactions between age and treatments led to differences in age-specific patterns of the fertility rate, which was highest in kairomone treatment in the first 50 days (figure 2*c*). Aggregated fertility data revealed a 43% higher overall fertility rate in kairomone than in control treatment ($z = 24.20$, $p < 0.0001$; figure 2*d*).

# 4. Discussion

We demonstrate that *Daphnia* of two species exposed to cues of predation risk that is decreasing with prey age and size live as long as the unexposed control animals. This contrasts the life-shortening effects that have been observed in a parallel study in the presence of fish cues, which indicate a predation risk that is increasing with age and size of the prey [33]. These contrasting effects are not immediately obvious in the light of the life-history theory based on the declining force of selection with age driven via the decreased probability of survival.

The simple 'classic' predictions of the theory state that under high adult predation risk, shorter lifespan or earlier or faster senescence should evolve [16] (but see [18–20]). The hypothesis may hold true in populations regulated via most abundant kinds of density dependence or affected by particular types of age- or stage-dependent mortality [19,20]. For example, when predation risk increases with size, such as in the case of visually oriented vertebrate predators feeding on zooplankton, increased early investments in reproduction may be coupled with a negligible cost of a physiologically shorter lifespan. Lifespan is reduced owing to decreased allocations to tissue repairs and maintenance. The survival cost of the reduction is negligible, because the risk of dying of extrinsic causes increases with gradually increasing size anyway (hence also with age). The short lifespan under such a risk is reported by both comparative studies of populations from habitats of different predation risk [59,60], as well as seen in phenotypic responses upon direct exposure to simulated predation [37,61] (both genetic and phenotypic effects discussed by Pietrzak *et al*. [33]).

In an opposite scenario, when an individual is gradually released from the mortality risk as it grows and ages, i.e. under predation risk decreasing with size, the optimal strategy should include fast growth, which enables it to quickly escape predator pressure. Such an intensive growth of the body consumes resources that could be invested in reproduction and tissue maintenance when the predator is absent. If less resources are available for soma maintenance, it seems to be clear that the effect of mortality risk that is decreasing with age on senescence should again support William's hypothesis [16].

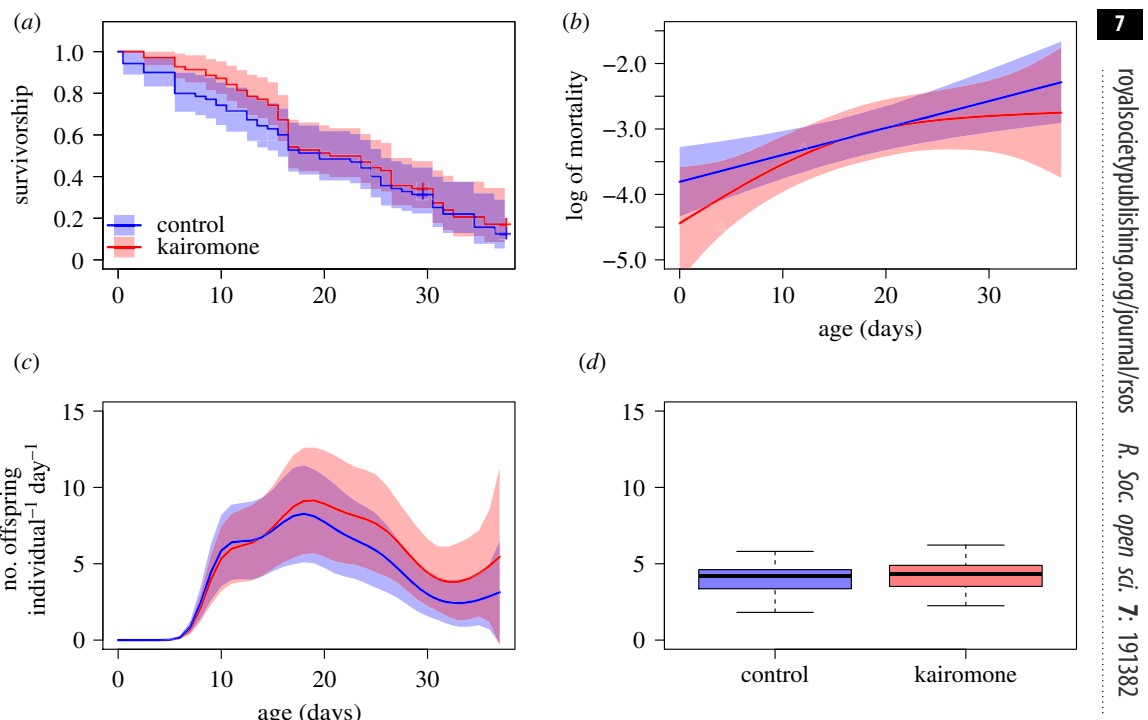

**Figure 1.** Fitness components of *D. pulex* either exposed to *Chaoborus* kairomone (red) or in control treatment (blue): (*a*) Kaplan–Meier estimator of survival, (*b*) age-specific mortality rate, (*c*) age-specific fertility rate and (*d*) overall fertility rate.

**Table 2.** *Chaoborus* experiment. (Summary of the fitted GAM (negative binomial) model to the mortality data.)

| fixed terms | estimate | s.e. | z-value | Pr(>\|z\|) |
|---|---|---|---|---|
| (intercept) | −3.1192 | 0.1661 | −18.78 | 0.0000 |
| treatment (*Chaoborus*) | −0.1843 | 0.2396 | 0.77 | 0.4420 |
| **smooth terms** | **Edf** | **Ref.df** | **$\chi^2$** | **$p(>\chi^2)$** |
| s(age) | 0.9070 | 9 | 9.95 | 0.0006 |
| ti(age) : treatment (control) | 0.0002 | 4 | 0.00 | 0.1333 |
| ti(age) : treatment (*Chaoborus*) | 0.8691 | 4 | 1.66 | 0.3970 |

**Table 3.** *Chaoborus* experiment. (Summary of the fitted GAM (negative binomial) model to the fertility data.)

| fixed terms | estimate | s.e. | z-value | Pr(>\|z\|) |
|---|---|---|---|---|
| (intercept) | −0.4499 | 0.2728 | −1.65 | 0.0992 |
| treatment (*Chaoborus*) | 0.0550 | 0.1432 | 0.37 | 0.7103 |
| **smooth terms** | **Edf** | **Ref.df** | **$\chi^2$** | **$p(>\chi^2)$** |
| s(age) | 7.8742 | 9 | 182.36 | 0.0000 |
| ti(age) : treatment (control) | 0.3516 | 9 | 0.53 | 0.0572 |
| ti(age) : treatment (*Chaoborus*) | 0.3850 | 9 | 0.61 | 0.0573 |

However, this may not be a general rule, especially if an organism is an indeterminate grower [36]. In the predation scenario considered here, older individuals are not only less affected by predation, i.e. survive better, but are also (gradually) more fecund, because clutch size typically increases with female body size (e.g. [62,63]). In such a case, later reproductive ages are important contributors to fitness, and hence, soma needs to be maintained in a good shape [36]. Although

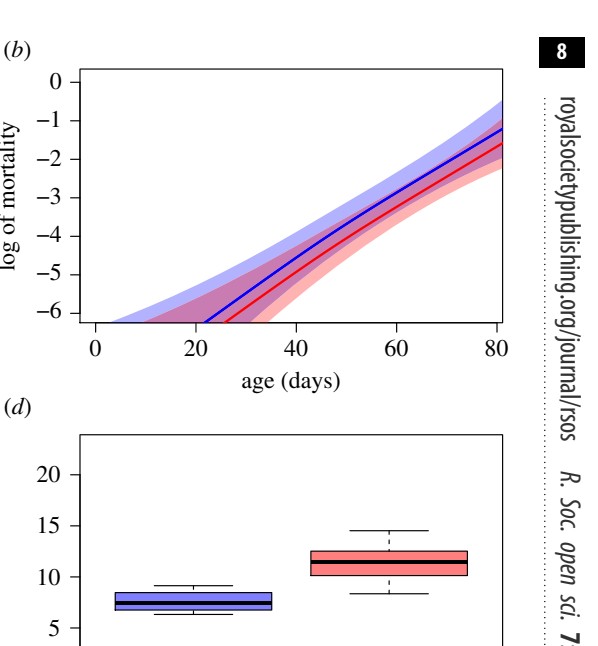

**Figure 2.** Fitness components of *D. magna* either exposed to *Triops* kairomone (red) or in control treatment (blue): (*a*) Kaplan–Meier estimator of survival, (*b*) age-specific mortality rate, (*c*) age-specific fertility rate and (*d*) overall fertility rate.

**Table 4.** *Triops* experiment. (Summary of the fitted GAM (negative binomial) model to the mortality data.)

| fixed terms | estimate | s.e. | $z$-value | $Pr(>|z|)$ |
| --- | --- | --- | --- | --- |
| (intercept) | −4.4536 | 0.3412 | −13.05 | 0.0000 |
| treatment (*Triops*) | −0.3714 | 0.3444 | −1.08 | 0.2809 |
| **smooth terms** | **Edf** | **Ref.df** | **$\chi^2$** | **$p(>\chi^2)$** |
| s(age) | 1.2414 | 19 | 52.51 | 0.0000 |
| ti(age) : treatment (control) | 0.0000 | 19 | 0.00 | 0.8316 |
| ti(age) : treatment (*Triops*) | 0.0000 | 19 | 0.00 | 0.7026 |

**Table 5.** *Triops* experiment. (Summary of the fitted GAM (negative binomial) model to the fertility data.)

| fixed terms | estimate | s.e. | $z$-value | $Pr(>|z|)$ |
| --- | --- | --- | --- | --- |
| (intercept) | 1.6866 | 0.0511 | 33.00 | 0.0000 |
| treatment (Induced) | 0.3963 | 0.0511 | 7.76 | 0.0000 |
| **smooth terms** | **Edf** | **Ref.df** | **$\chi^2$** | **$p(>\chi^2)$** |
| s(age) | 15.3850 | 19 | 156.20 | 0.0000 |
| ti(age) : treatment (control) | 0.1752 | 19 | 0.18 | 0.0000 |
| ti(age) : treatment (*Triops*) | 6.8754 | 19 | 43.24 | 0.0000 |

there are reports that simulating such predation regime have no effects on lifespan [64,65], these effects have not been explicitly approached and discussed in the context of age-dependent predation pressure.

Further, the adaptive phenotypic responses to the perceived threat can be expected to be parallel, to an extent, to the evolutionary outcomes of the actual mortality applied [33,66,67]. The latter is well exemplified by human exploitation or purposeful artificial selection in wild or captive populations. Size and age selective fishing induces the evolution of early maturity, slower growth, smaller size [68–70]. Early studies in insects showed that directly selecting for further culture, only offspring born to primiparous parents lead to the quick evolution of reduced lifespan [71]. On the other end, selection for late reproduction, which is not equivalent to predation on young adults, but in ways similar, leads to increased longevity [72]. *Drosophila* lifespan increased after three generations of thus introduced selection [73].

The chemical nature of predator cues perceived by prey is not known in most systems; yet, recent advances shed new light on both general principles and particular systems [74–76]. This is important as the nature of the cue and its predator specificity may present a constraint for recognition. Most likely different predators are recognized by different mixtures of chemicals released. This has recently been suggested by studies targeting *Daphnia* as prey. While a group of lipidated glutamine conjugates released by *Chaoborus* larvae induces morphological defence in *D. pulex* [77], a single bile salt released by fish, 5α-cyprinol sulfate, induces defence behaviour in *D. magna* [78].

We hypothesized that in naturally evolved prey–predator systems, the prey can perceive the specific identity of the predator, which is coupled to a particular pattern of age and size-specific mortality, and thus perceive the nuances of size-dependence of the threat and modulate its life history accordingly. We confirmed that under predation risk decreasing with size in an indeterminate grower, no acceleration of senescence was observed. Our results stay in accordance with the main prediction—we observed no effects on mortality patterns of perceived predation risk that is decreasing with prey size and age, in any of the two studied systems. It is particularly noteworthy in the case of *D. magna* exposed to *Triops* presence, as the exposed animals live as long as those unexposed, despite the increased fecundity and the well-documented investments in morphological defence [48,50]. The higher fertility rates in the kairomone treatment compared to the control treatment may be related to the larger body size of the kairomone-exposed females [48]. They might also result from higher food supply associated with bacteria growth on predator exudates [79] not eliminated in the *Triops* study. Anyway, all, faster growth, armour and higher reproductive investment are expected to implicate costs, which, in this study, were not observed (see [80] for a parallel case). We might speculate on how the kairomone-induced adaptive morphology and life-history shifts are traded-off in the real environmental setting, for instance, larger animals may be more susceptible to visual hunting predators (environmental costs). However, of particular interest to this study, performed in safe laboratory conditions, is that no survival reduction was observed. Indeed, according to the main hypothesis, late survival should be here under strong selection: the largest individuals are least vulnerable to predation and have the highest reproductive potential.

For the first time, we directly address the phenomenon of an organism modulating its life-history course, senescence and lifespan in particular, according to the size- and age-dependence of the predation threat perceived. Combined with previously published data, our results suggest *Daphnia* can discriminate between predation risk that decreases and one that increases with size and age, and modulate its life history accordingly. The next step will be to test the effects of the perceived threats of both regimes simultaneously in a single species, on one hand, and to perform a large comparative study of different prey–predator systems, on the other. Further research, both theoretical and empirical, is much needed.

Data accessibility. Data is available from the Dryad Digital Repository: https://doi.org/10.5061/dryad.gb89p60 [81].

Authors' contributions. B.P. conceived the idea and led writing of the manuscript. B.P., M.Ra., M.Re. and C.L. collected and provided the data. M.J.D. analysed the data and cooperated in developing the idea and writing. All authors contributed critically to the drafts or revision and gave final approval for publication.

Competing interests. We declare we have no competing interests.

Funding. This study was partially supported by and realized within statutory research of the University of Warsaw.

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
