## [Reviewer comments · Royal Society Open Science]

Review History

RSOS-191382.R0 (Original submission)

Review form: Reviewer 1

Is the manuscript scientifically sound in its present form?

Yes

Are the interpretations and conclusions justified by the results?

Yes

Is the language acceptable?

Yes

Do you have any ethical concerns with this paper?

No

Have you any concerns about statistical analyses in this paper?

No

Recommendation?

Accept with minor revision (please list in comments)

Comments to the Author(s)

This is a really nice paper on an important theme. The paper is well grounded and provides appropriate context. I see no real issues in the experiments or analysis. The discussion does a good job of discussing the underlying general issues and provides some potentially interesting paths forward.

As per my comment on page 18 (last para), my only concern is for the authors to think a little bit more about the potential chemical cue information available to prey, which could well affect their responses to suites of predators. There is not a lot of information out there on this, but what is there suggests that the nature of the cues used by prey to discriminate predators and their risk might constrain subsequent prey responses.

There are a few grammatical and stylistic suggestions/corrections on the ms (Appendix A), but all of them are minor. The paper is by and large very well written.

Review form: Reviewer 2

Is the manuscript scientifically sound in its present form?

Yes

Are the interpretations and conclusions justified by the results?

No

Is the language acceptable?

Yes

Do you have any ethical concerns with this paper?

No

Have you any concerns about statistical analyses in this paper?

Yes

Recommendation?

Accept with minor revision (please list in comments)

Comments to the Author(s)

Pietrzak et al aimed to test the hypothesis that under predation risk that decreases with prey size and age, senescence deceleration may occur in the prey. They find no evidence of such deceleration. Negative results should be published with the same priority as those that corroborate the predictions of a study. Therefore this paper should be published, with some amendments that improve discussion.

I would question the premise that predation decreasing with age should necessarily lead to decreased senescence. As the authors correctly indicate in the Discussion (lines 284-290), if senescence is a result of life-history trade-offs, then, just as early investments in reproduction are expected to increase senescence in case of predation increasing with age, early investment into somatic growth, at the expense of maintenance and repair, might do the same. Authors should reformulate their predictions in the light of this reasoning.

Minor comments:

55 (and 271): increased extrinsic mortality leads to potential lifespan shortening via increased rates of senescence (but see [17-20]) -- increased extrinsic mortality does not leads to increased rates of senescence (as outlined in ref.20) – the increase of mortality with age does.

Fig. 1b and 2b: log-mortality is a noisy estimate, particularly with relatively small cohorts. The authors should present raw data points rather than just replacing them with regression lines.

Tables 2 - 4: terms $t_i(\text{age})$ must be explained. The text mentions t_s terms – are these the same?

Decision letter (RSOS-191382.R0)

14-Nov-2019

Dear Dr Pietrzak,

On behalf of the Editors, I am pleased to inform you that your Manuscript RSOS-191382 entitled "Phenotypic plasticity of senescence in *Daphnia* under predation impact: No ageing acceleration when the perceived risk decreases with age" has been accepted for publication in Royal Society Open Science subject to minor revision in accordance with the referee suggestions. Please find the referees' comments at the end of this email.

The reviewers and handling editors have recommended publication, but also suggest some minor revisions to your manuscript. Therefore, I invite you to respond to the comments and revise your manuscript.

- Ethics statement

- Data accessibility

<http://datadryad.org/submit?journalID=RSOS&manu=RSOS-191382>

- Competing interests

- Authors' contributions

- Acknowledgements

- Funding statement

Because the schedule for publication is very tight, it is a condition of publication that you submit the revised version of your manuscript before 23-Nov-2019. Please note that the revision deadline will expire at 00.00am on this date. If you do not think you will be able to meet this date please let me know immediately.

- 1) A text file of the manuscript (tex, txt, rtf, docx or doc), references, tables (including captions) and figure captions. Do not upload a PDF as your "Main Document";

- 2) A separate electronic file of each figure (EPS or print-quality PDF preferred (either format should be produced directly from original creation package), or original software format);
- 3) Included a 100 word media summary of your paper when requested at submission. Please ensure you have entered correct contact details (email, institution and telephone) in your user account;
- 4) Included the raw data to support the claims made in your paper. You can either include your data as electronic supplementary material or upload to a repository and include the relevant doi within your manuscript. Make sure it is clear in your data accessibility statement how the data can be accessed;
- 5) All supplementary materials accompanying an accepted article will be treated as in their final form. Note that the Royal Society will neither edit nor typeset supplementary material and it will be hosted as provided. Please ensure that the supplementary material includes the paper details where possible (authors, article title, journal name).

If your manuscript is newly submitted and subsequently accepted for publication, you will be asked to pay the article processing charge, unless you request a waiver and this is approved by Royal Society Publishing. You can find out more about the charges at <https://royalsocietypublishing.org/rsos/charges>. Should you have any queries, please contact openscience@royalsociety.org.

Kind regards,

on behalf of the Associate Editor and Professor Kevin Padian (Subject Editor)
openscience@royalsociety.org

Associate Editor Comments to Author:

Please ensure you fully respond to the queries presented by the reviewers, and include a full point-by-point response to these with your revision.

Reviewer comments to Author:

Reviewer: 1

Comments to the Author(s)

This is a really nice paper on an important theme. The paper is well grounded and provides appropriate context. I see no real issues in the experiments or analysis. The discussion does a good job of discussing the underlying general issues and provides some potentially interesting paths forward.

As per my comment on page 18 (last para), my only concern is for the authors to think a little bit more about the potential chemical cue information available to prey, which could well affect their responses to suites of predators. There is not a lot of information out there on this, but what is there suggests that the nature of the cues used by prey to discriminate predators and their risk might constrain subsequent prey responses.

There are a few grammatical and stylistic suggestions/corrections on the ms, but all of them are minor. The paper is by and large very well written.

Reviewer: 2

Comments to the Author(s)

Pietrzak et al aimed to test the hypothesis that under predation risk that decreases with prey size and age, senescence deceleration may occur in the prey. They find no evidence of such deceleration. Negative results should be published with the same priority as those that corroborate the predictions of a study. Therefore this paper should be published, with some amendments that improve discussion.

I would question the premise that predation decreasing with age should necessarily lead to decreased senescence. As the authors correctly indicate in the Discussion (lines 284-290), if senescence is a result of life-history trade-offs, then, just as early investments in reproduction are expected to increase senescence in case of predation increasing with age, early investment into somatic growth, at the expense of maintenance and repair, might do the same. Authors should reformulate their predictions in the light of this reasoning.

Minor comments:

55 (and 271): increased extrinsic mortality leads to potential lifespan shortening via increased rates of senescence (but see [17-20]) -- increased extrinsic mortality does not lead to increased rates of senescence (as outlined in ref.20) – the increase of mortality with age does.

Fig. 1b and 2b: log-mortality is a noisy estimate, particularly with relatively small cohorts. The authors should present raw data points rather than just replacing them with regression lines.

Tables 2 - 4: terms $t_i(\text{age})$ must be explained. The text mentions t_s terms – are these the same?

Author's Response to Decision Letter for (RSOS-191382.R0)

See Appendix B.

Decision letter (RSOS-191382.R1)

19-Dec-2019

Dear Dr Pietrzak,

It is a pleasure to accept your manuscript entitled "Phenotypic plasticity of senescence in *Daphnia* under predation impact: No ageing acceleration when the perceived risk decreases with age" in its current form for publication in Royal Society Open Science.

Best regards,

on behalf of the Associate Editor and Professor Kevin Padian (Subject Editor)
openscience@royalsociety.org

Appendix A**ROYAL SOCIETY
OPEN SCIENCE****Phenotypic plasticity of senescence in *Daphnia* under predation impact: No ageing acceleration when the perceived risk decreases with age**

Journal:	Royal Society Open Science
Manuscript ID	RSOS-191382
Article Type:	Research
Date Submitted by the Author:	26-Sep-2019
Complete List of Authors:	Pietrzak, Barbara; University of Warsaw, Faculty of Biology, Department of Hydrobiology Rabus, Max; University of Bayreuth, Animal Ecology I Religa, Maciej; University of Warsaw, Faculty of Biology, Department of Hydrobiology Laforsch, Christian; University of Bayreuth, Animal Ecology I Danko , Maciej; Max-Planck-Institute for Demographic Research
Subject:	ecology < BIOLOGY, evolution < BIOLOGY
Keywords:	phenotypic plasticity, antipredator response, kairomone, lifespan, longevity, senescence
Subject Category:	Biology (whole organism)

Author-supplied statements

Relevant information will appear here if provided.

Ethics

Does your article include research that required ethical approval or permits?:

This article does not present research with ethical considerations

Statement (if applicable):

CUST_IF_YES_ETHICS :No data available.

Data

It is a condition of publication that data, code and materials supporting your paper are made publicly available. Does your paper present new data?:

Yes

Statement (if applicable):

Data available from the Dryad Digital Repository: doi:10.5061/dryad.gb89p60, URL link:

<https://datadryad.org/review?doi=doi:10.5061/dryad.gb89p60>

Conflict of interest

I/We declare we have no competing interests

Statement (if applicable):

CUST_STATE_CONFLICT :No data available.

Authors' contributions

This paper has multiple authors and our individual contributions were as below

Statement (if applicable):

B.P. conceived the idea and led writing the manuscript. B.P., M.Ra., M.Re. and C.L. collected and provided the data. M.J.D. analysed the data and cooperated in developing the idea and writing.

M.Ra. and C.L. contributed critically to the drafts. All authors gave final approval for publication.

1
2

[revised manuscript text omitted]

347

42 348 **Data**

349 Data available from the Dryad Digital Repository: doi:10.5061/dryad.gb89p60, URL link:

350 <https://datadryad.org/review?doi=doi:10.5061/dryad.gb89p60>

351

52 352 **Author contributions**

353 B.P. conceived the idea and led writing the manuscript. B.P., M.Ra., M.Re. and C.L. collected and

354 provided the data. M.J.D. analysed the data and cooperated in developing the idea and writing.

355 M.Ra. and C.L. contributed critically to the drafts. All authors gave final approval for

publication.

[revised manuscript text omitted]

Appendix B

We thank both Reviewers, their comments helped us improve the manuscript. We now stated our predictions more clearly, rephrased some parts of the text, clarified methods and raised the topic of potential constraints in recognizing predators, all in accordance with the suggestions.

Below you will find detailed response to the all the comments, our reply in blue font. We submit a track-changes word file where the introduced changes can be seen. We also submit a clean version with changes accepted and figures moved to separate files.

Reviewer comments to Author:

Reviewer: 1

This is a really nice paper on an important theme. The paper is well grounded and provides appropriate context. I see no real issues in the experiments or analysis. The discussion does a good job of discussing the underlying general issues and provides some potentially interesting paths forward.

As per my comment on page 18 (last para), my only concern is for the authors to think a little bit more about the potential chemical cue information available to prey, which could well affect their responses to suites of predators. There is not a lot of information out there on this, but what is there suggests that the nature of the cues used by prey to discriminate predators and their risk might constrain subsequent prey responses.

[comment in text] Although this might be beyond the scope of the present contribution, one does need to be cognizant of potential constraints. It's unclear what lends specificity to predator cues; the differences can be either qualitative or quantitative (Poulin et al., PNAS, Weissburg et al J Chem Ecol both in the last 3 years. Quant differences (where cues are characterized by the same molecules at differing amounts) mean prey might not be able to accurately discern the blend as a combination of two distinct predator cues. This would only be possible if different molecules indicate the predator type.

Thank you for bringing up this important topic, which we did not refer to before. We now added a short paragraph raising this topic and pointing to very recent studies showing different molecules released by midge larvae and fish are recognized by Daphnia.

There a few grammatical and stylistic suggestions/corrections on the ms, but all of them are minor. The paper is by and large very well written.

We correct the text according to suggestions.

Reviewer: 2

Pietrzak et al aimed to test the hypothesis that under predation risk that decreases with prey size and age, senescence deceleration may occur in the prey. They find no evidence of such deceleration. Negative results should be published with the same priority as those that corroborate the predictions of a study. Therefore this paper should be published, with some amendments that improve discussion.

I would question the premise that predation decreasing with age should necessarily lead to decreased senescence. As the authors correctly indicate in the Discussion (lines 284-290), if

senescence is a result of life-history trade-offs, then, just as early investments in reproduction are expected to increase senescence in case of predation increasing with age, early investment into somatic growth, at the expense of maintenance and repair, might do the same. Authors should reformulate their predictions in the light of this reasoning.

Thank you for this comment, we agree that faster senescence is a plausible scenario under growth-inducing predation risk that decreases with prey age. We tackle this important issue before introducing our predictions – living in stress and inducing growth or defences might bring costs later in life, yet, with continuous growth, associated with increasing fecundity and tissue renewal, these costs may be overthrown [lines 79-88]. We clarify now our reasoning in line with your suggestion by specifying when our predictions are most likely to be met [lines 94-96]. To keep the text focused, we leave further considerations and alternative scenarios for discussion.

Minor comments:

55 (and 271): increased extrinsic mortality leads to potential lifespan shortening via increased rates of senescence (but see [17-20]) -- increased extrinsic mortality does not lead to increased rates of senescence (as outlined in ref.20) – the increase of mortality with age does.

Thank you, we were too brief in presenting and contrasting the early simple prediction – “classic” prediction by large and long functioning in this form in literature – and its later more specific and correct formulation. We clarify this now according to your suggestion [lines 54-58]. In discussion, we develop in the paragraph specifying under which circumstances and conditions extrinsic mortality leads to increased rates of senescence (density dependent or age dependent mortality) [lines 280-292].

Fig. 1b and 2b: log-mortality is a noisy estimate, particularly with relatively small cohorts. The authors should present raw data points rather than just replacing them with regression lines.

We considered presenting raw data points, yet decide to keep the figures as they are presented now for the following reasons.

We do not use log-mortality as an estimator, as we fitted mortality models using (restricted) maximum likelihood approach within Generalized Additive Models. We use death counts as dependent variable, log of exposures as offset. The log-mortality is used only to plot mortality curves. The log scale is typically used for plotting mortality, particularly if the mortality increases exponentially with age.

Presenting "raw" data points is very problematic for mortality measured in very short intervals and with moderate sample sizes (small number of events within an age interval – and “raw” is in our case data for age intervals of length of one day). Typically then and also in our case, there is an excess of points laying on OX axis, which makes interpretation of the age-specific pattern difficult. There were attempts to fit ordinary least squares models to such "raw" mortality data in the past, yet this was not correct. Currently, adding mortality points (and using them to fit the models) is not a common practise, e.g. Jones et al. 2014 (Nature 505: 169-173). Alternatively, such points can be plotted for aggregated intervals (e.g. [33] Pietrzak et al. 2015 (Exp. Gerontol.)), but then they are no longer "raw".

You will find a sample plot of “raw” data points at the end of this response (Chaoborus experiment, age in days, closed and open circles for control and predator treatments, respectively).

Tables 2 - 4: terms $t_i(\text{age})$ must be explained. The text mentions t_s terms – are these the same?

We agree this was not properly stated, we improved the description in "data analysis" now.

Reviewers' comments in text:

56 (58 now in ms with changes tracked): *gave also ground for* ""laid the ground work for""
corrected

63 (65): *mortality without age pattern* ""age-independent predator mortality"" corrected

87 (90): *the* deleted

149-151 (154-157): *Kairomone water was obtained by incubating Chaoborus larvae at a density of 25 ind. 150 L⁻¹ for 48 h, at 4 °C to prevent quick development and metamorphosis during the experiment, and fed with D. pulex.* ""okay-how is it clear that cue quality/quantity was maintained? This will be dependent on predator state and predation. It would be helpful to have some indication (for both predators) that these remained constant. Although that might be hard for predator state (at least quantitatively), one might be able to make more detailed statements about predation rate.""

Thank you for this comment. We fully agree, it is crucial to maintain constant conditions for predator incubation throughout experiment. That is how we kept both of them and now clarify this in the text.

(274): *which* "that" corrected

(275): *size,* ""remove the comma after this word"" corrected

(276-278): [33] ""I do not know this study, but one has to be careful that it is parallel and involves studies with a similar time course of exposure.""

Thank you, this is important to point to the fact these studies were indeed parallel and we do this now. The cited study, which two of us co-authored, followed the same principle and time course of exposure: whole life exposure to constant predator cues. Moreover, the Chaoborus-part experimental setup here was designed to mimic that of the cited study.

(302): *here considered predation scenario* ""predation scenario considered here"" corrected

(310): *to an extent* ""there should be commas before and after this phrase"" corrected

308-310 (326-328): *We hypothesized that in naturally evolved prey-predator systems the prey can perceive the nuances of size-dependence of the perceived threat and modulate its life history accordingly,* ""what is likely being perceived is the specific identity of the predator, which is coupled to a particular pattern of age/size specific mortality. The statement here is not incorrect, but might not be interpreted properly by the intended audience. Smeets has a nice recent review on chemical perception of predators by prey""

We clarify this statement now for more proper interpretation.